# Mapping Floristic Patterns of Trees in Peruvian Amazonia Using Remote Sensing and Machine Learning

**Pablo Pérez Chaves [1,]*** , **Gabriela Zuquim [1,2]** , **Kalle Ruokolainen [1]** , **Jasper Van doninck [3]** , **Risto Kalliola [3]** , **Elvira Gómez Rivero [4] and Hanna Tuomisto [1]**

[1] Department of Biology, University of Turku, 20014 Turku, Finland; gabzuq@utu.fi (G.Z.); kalle.ruokolainen@utu.fi (K.R.); hanna.tuomisto@utu.fi (H.T.)
[2] Department of Biology, Aarhus University, Nordre Ringgade 1, DK-8000 Aarhus, Denmark
[3] Department of Geography and Geology, University of Turku, 20014 Turku, Finland; jasper.vandoninck@utu.fi (J.V.d.); riskall@utu.fi (R.K.)
[4] Servicio Forestal Nacional y de Fauna Silvestre, Lima 15076, Peru; egomez@serfor.gob.pe
* Correspondence: papech@utu.fi

**Abstract:** Recognition of the spatial variation in tree species composition is a necessary precondition for wise management and conservation of forests. In the Peruvian Amazonia, this goal is not yet achieved mostly because adequate species inventory data has been lacking. The recently started Peruvian national forest inventory (INFFS) is expected to change the situation. Here, we analyzed genus-level variation, summarized through non-metric multidimensional scaling (NMDS), in a set of 157 INFFS inventory plots in lowland to low mountain rain forests (<2000 m above sea level) using Landsat satellite imagery and climatic, edaphic, and elevation data as predictor variables. Genus-level floristic patterns have earlier been found to be indicative of species-level patterns. In correlation tests, the floristic variation of tree genera was most strongly related to Landsat variables and secondly to climatic variables. We used random forest regression, under varying criteria of feature selection and cross-validation, to predict the floristic composition on the basis of Landsat and environmental data. The best model explained >60% of the variation along NMDS axes 1 and 2 and 40% of the variation along NMDS axis 3. We used this model to predict the three NMDS dimensions at a 450-m resolution over all of the Peruvian Amazonia and classified the pixels into 10 floristic classes using k-means classification. An indicator analysis identified statistically significant indicator genera for 8 out of the 10 classes. The results are congruent with earlier studies, suggesting that the approach is robust and can be applied to other tropical regions, which is useful for reducing research gaps and for identifying suitable areas for conservation.

**Keywords:** tropical forests; biogeography; community composition; forest classification; Landsat; random forest; forest inventory; Peru

## 1. Introduction

Deforestation, forest fires, habitat loss, and climate change are continuous threats in Amazonia, the largest and most biologically diverse tropical forest in the world. At least 17% of Amazonia has already been deforested, especially in the border of the biome [1], while vast areas are still botanically poorly known [2–4]. Field data are essential for conservation planning, but data collection is time consuming and costly, while deforestation occurs at a fast pace.

Remote sensing data provide spatially and temporally continuous information useful for biodiversity assessments [5–7]. Therefore, combining existing biological field data with remote

sensing is the method of choice to obtain an overview and to make spatial predictions of various biological patterns over large areas [5–14]. This can significantly improve the information needed for land use planning, reduce knowledge gaps, and help to identify priority areas for the conservation of Amazonian biodiversity.

Several studies have taken advantage of remotely sensed data to predict spatial patterns in different aspects of biological diversity in Amazonia [9–13,15–18]. Early studies used visual interpretation to identify different vegetation types [19], but increased data availability and improved analytical methods now make more sophisticated analyses possible. For example, functional and chemical traits of the forest canopy and carbon stocks of vegetation have been mapped in many parts of Amazonia using airborne and satellite sensors [9–12,20–22].

For conservation purposes, it is especially important to have information on species occurrences and community composition, as well as on how these vary in space. Making spatially explicit predictive models of these variables requires an understating on how the community composition is related to such environmental factors or other predictors that are available as raster layers over the area of interest. The accumulated body of evidence from ecological studies suggests that edaphic and climatic variables are good predictors of the plant species distributions and community composition in Amazonia. Evidence on this has been especially accumulated for understory plants, such as ferns and Melastomataceae [13,23–27], Zingiberales [28], and palms [29–31], but also for canopy trees [17,32–40]. In addition, it has been found that the compositional turnover of trees is strongly correlated with understory plant groups [19,24,36,41,42], so results on the latter can be used as an indication of what to expect with the former. Accordingly, it is reasonable to expect that it is possible to predict the floristic turnover of trees in Amazonia on the basis of environmental variables. A large number of environmental data layers are already freely available and can be used for this purpose, although they still suffer from accuracy problems [43,44].

When compared to environmental data layers that are based on interpolating among (sometimes very sparse) field-verified measurements, remotely sensed data have the great advantage that they provide consistent data over large areas. Several studies have explored the ability of Landsat imagery to predict spatial patterns in soil properties or the understory species composition at the landscape to regional scales [24–26,45–48], and a few have done this across the Amazon [13,49]. The results have indicated that Landsat images successfully reflect such environmental variation that is relevant for Amazonian plants, and that this appears mostly to be related to edaphic conditions, even at the Amazon-wide extent [13,49]. Consistently with these results, satellite images have also been successfully used as predictive data layers in the species distribution modelling of ferns [50] and trees [14,18]. Therefore, freely available remote sensing data, such as Landsat images, are potentially very useful for mapping the floristic composition patterns of Amazonian trees.

The bottleneck is floristic field data. To properly map floristic variation in large areas, such as the whole Peruvian Amazonia, a large number of systematic, comparable, and well-distributed field surveys are required. The Peruvian national forest inventory aims to establish plots with such characteristics in order to characterize forest resources. Thus far, accurate and cross-checked species-level identifications are not yet available, but many inventory plots have already been established throughout Peruvian Amazonia and genus-level identifications are already available. Therefore, it is possible to combine a unique set of field surveys and remote sensing and environmental layers to investigate the floristic patterns of trees in Peruvian Amazonia at the genus level.

The use of genus-level data has been a common approach in studies focusing on the community composition of Amazonian trees. For instance, gradients in tree composition were modelled across all Amazonia using spatial interpolation of genus-level data [17], and relationships between tree genus turnover and environmental differences were assessed [51]. Studies that have specifically compared the use of genus-level and species-level data have found a high degree of congruence, and that although upscaling to genus resolutions leads to some loss of ecological information, the most important patterns have been captured [51–53].

In this study, we were interested in obtaining a map of the genus-level compositional patterns of trees in Peruvian Amazonia using satellite imagery and environmental data. For understory plants, such maps have been produced for a small area in Ecuadorian Amazonia [48] and for the entire Amazon biome [13]. For trees, the floristic composition at the genus level has been modelled across the Amazon biome by spatial interpolation, but without the use of environmental or remote sensing data [17].

To achieve such a map, we first determined the main tree floristic gradients at the genus level across Peruvian Amazonia and their correlates with Landsat data and environmental layers. We then spatially predicted each of the three axes of a floristic ordination using random forest regression and Landsat data and environmental variables in order to derive the first floristic map of Peruvian Amazonia. We further classified the floristic map and ran an indicator genus analysis to identify which genera were affiliated to each of the classes. We addressed these questions using tree field data from the Peruvian National Forest and Wildlife Inventory (Inventario Nacional Forestal y de Fauna Silvestre, INFFS) of the National Forest and Wildlife Service (Servicio Nacional Forestal y de Fauna Silvestre, SERFOR), consisting of 157 tree inventory plots distributed among Peruvian Amazonia.

Our study had three main aims: (1) To clarify how the floristic patterns of tree genera across Peruvian Amazonia are related with canopy reflectance and available environmental data layers; (2) to visualize genus-level floristic variation of trees across Peruvian Amazonia; and (3) to derive a classification of the genus-level floristic variation and to assess if the recognized classes have characteristic indicator genera.

## 2. Materials and Methods

### 2.1. Study Area

The study area comprehends all Peruvian Amazonia as defined by the Peruvian Ministry of Environment [54], covering approximately 790,000 km$^2$ (Figure 1a). The climate is mainly tropical and humid with a monthly average temperature of 23.1 °C (ranging between 4.3 and 26.2 °C) and an average annual rainfall of 2314 mm (ranging from 400 to 8000 mm) as extracted from Climatologies at High Resolution for the Earth's Land Surface Areas CHELSA [55] (Figure 1b,c). The elevation ranges between 100 and 3000 m above sea level (Figure 1d), but here, we limited the analyses to areas below 2000 m on the basis of the field data coverage. Soils tend to be richer closer to the Andes (Figure 1e).

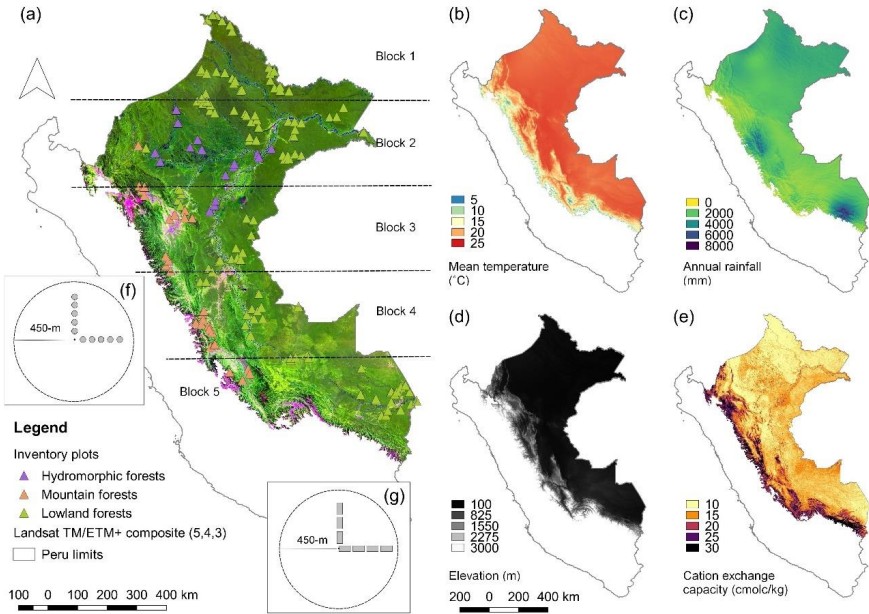

**Figure 1.** Geographical characteristics of the study area in Peruvian Amazonia. (**a**) Landsat TM/ETM+ composite (bands 5, 4, and 3 assigned to red, green, and blue, respectively) of Peruvian Amazonia



with the sampling setup of the inventory plots and the five latitudinally defined blocks used in cross-validation of the models, (**b**) Mean annual temperature, (**c**) annual precipitation, (**d**) elevation, and (**e**) cation exchange capacity (CEC) in the study area. (**f**) Each inventory plot in hydromorphic (16 plots) or mountain forests (28 plots) consisted of 10 round subunits of 0.05 ha separated by 30 m and (**g**) in lowland forests (113 plots) of 7 rectangular subunits of 0.1 ha (20 m × 50 m) separated by 75 m. The circle with a radius of 450 m in (**f**) and (**g**) depicts the buffer we used for extracting Landsat and elevation data. Panels (**g**) and (**f**) are schemes not drawn to scale.

## 2.2. Floristic Data

Floristic data were collected between 2013 and 2018 in 157 sampling plots well distributed across Peruvian Amazonia (Figure 1a). The data were provided by SERFOR, which is part of the INFFS. One of the objectives of the INFFS is to evaluate forest resources throughout Peru in a multi-stage process that is planned to eventually include over 1800 inventory plots. We used inventory data from the first of five so-called panels, since it has already been completed and processed. Data were available from various ecozones, but we excluded the data from outside Amazonia (Andes and the coast) and focused on lowland forests, mountain forests, and hydromorphic forests. The definition criteria for the ecozones are listed in the descriptive memory of the ecozone map [56].

The inventory plots were L-shaped and consisted of either 10 0.05-ha subunits totaling 0.5 ha (hydromorphic or mountain forests) or seven 0.1-ha subunits totaling 0.7 ha (lowland forests; Figure 1f,g). Subunits in the 113 lowland forest plots were rectangular (20 m × 50 m) and separated from each other by a distance of 75 m. Subunits in the 16 hydromorphic forest plots and the 28 mountain forest plots were circular (12.62 m radius) and separated from each other by 30 m. The corner of each L-shaped plot was georeferenced using a Global Positioning System (GPS) device. Field sampling followed the protocols and guidelines of the INFFS methodological framework [57].

Trees (including palms and tree ferns) exceeding 30 cm in diameter at breast height (dbh) were recorded in all subunits of each plot. In addition, trees between 10 and 30 cm dbh were recorded in half of the subunits per plot. For each tree, its dbh (cm), height (m), scientific name, and local name were recorded in the field. Different field teams collected the inventory data and each team had a trained botanist who identified the trees with a field name and collected at least one representative voucher for each field name per plot. The voucher specimens were deposited in La Molina National Agrarian University Herbarium (MOL). Each botanist carried out the species identification of his/her voucher specimens individually, and voucher specimens were not cross-checked among botanists. Therefore, we used only genus-level identifications in the analyses even when species-level identifications were available in the inventory database.

## 2.3. Landsat and Environmental Data

Because Peruvian Amazonia has a moist climate, cloud-free satellite images are rare, and mosaicking Landsat imagery to obtain a seamless product is challenging. These problems have been overcome in a Landsat TM/ETM+ composite based on all Landsat acquisitions from the dry season months of the 10-year period 2000–2009 [49]. This time period was chosen because both Landsat 5 and 7 were operative at the time, which maximized data availability, thereby increasing the signal-to-noise ratio of the product [49]. When producing the composite, the Landsat images were corrected for atmospheric and directional effects. They were composited using the medoid method, which ensures that the reflectance values represent relatively stable ground cover characteristics that can be expected to be relevant for canopy trees. Details of the methodology are described in [49,58–60]. We cropped the composite to the extent of Peruvian Amazonia for use in the present study and masked out non-forested areas using an updated forest loss layer in the period 2001–2018 obtained from the Peruvian Ministry of Environment (http://geobosques.minam.gob.pe).

The Landsat TM/ETM+ composite consists of six bands that correspond to different sections of the electromagnetic spectrum: 1 (blue spectrum), 2 (green), 3 (red), 4 (near infrared), 5, and 7 (the latter

two shortwave infrared). Reflectance values from each of the Landsat bands were acquired at a 30-m spatial resolution and are collectively referred to as "Landsat data" hereafter.

We obtained elevation from the SRTM digital elevation model (Shuttle Radar Topography Mission) at a 30-m spatial resolution (DOI:/10.5066/F7PR7TFT). Soil cation exchange capacity (CEC) at a 0.05-m depth was obtained from SoilGrids [61] at a 250-m spatial resolution. Climatic data (19 bioclimatic variables) at approximately a 1-km spatial resolution were obtained from CHELSA [55]. Elevation, soil, and climatic data are hereafter collectively referred to as "environmental data". The names and abbreviations of all Landsat and environmental variables are listed in the Supporting Material (Table S1).

Climate and soil data were obtained for each inventory plot by extracting the required variable value from the pixel corresponding to the inventory plot coordinates. Because the Landsat and elevation data had a much higher spatial resolution, first a buffer of 450 m was defined around each inventory plot, and then the median value of the pixels within the buffer was extracted for each variable (Figure 1f,g). This served to reduce the effect of local-scale variability and to provide a more representative estimate of the site properties for regional comparisons.

### 2.4. Data Analyses

Our aim was to model the overall floristic patterns of trees at the community level. As tropical forests typically have hundreds of tree genera, many of which are of very low prevalence, and it is not practical to model each genus separately. Therefore, we first used an ordination (non-metric multidimensional scaling or NMDS) to summarize the main floristic variation among the forest inventory plots into three orthogonal axes (NMDS 1-3) of variation. As input data, we used a matrix of pairwise floristic dissimilarities between the plots. We used the Sørensen dissimilarity index, which uses only presence-absence data of the genera. The extended (step-across) version of the index was used, because it provides ecologically realistic dissimilarity values between plots that share no genera [27,62,63]. The NMDS results were used both to visualize the floristic dissimilarity patterns and to assess their relationship with potential explanatory variables. Linear regression analysis was used to quantify the relationship of each of the first three NMDS axes (NMDS 1–3) with the reflectance values of Landsat bands and with the environmental variables (elevation, bioclimatic data, and soil data).

Environmental and Landsat distance matrices were obtained using pairwise Euclidean distances between plots, calculated for each variable separately. This produced a total of 6 Landsat matrices (one per band) and 21 environmental matrices (19 bioclimatic variables, CEC, and elevation). Geographical distances between plots were calculated using plot coordinates and the values were transformed to their natural logarithm before analysis. Mantel tests of the matrix correspondence [64] were used to define whether floristic differences were correlated with Landsat and environmental distances. The Pearson correlation method was used in the test with 999 permutations. Partial Mantel tests were used to assess residual correlations after controlling for the effect of geographic distances.

To model the floristic patterns of trees across Peruvian Amazonia, we derived random forest (RF) regressions using each of the NMDS axes 1–3 as response variables and the Landsat and environmental variables as predictors. Different combinations of variables were used in the modeling procedure: (1) Landsat variables only, (2) environmental variables only, (3) all Landsat and environmental variables together, and (4) a selected combination of Landsat and environmental variables. Up to 500 random trees per NMDS axis were generated as separate models and were further averaged for the final prediction until there was no further increase in performance. To account for the correlation between random trees, we ran models with different combinations and numbers of predictors, ranging from two to the total number of potential predictor variables [65]. The best final model for each of the NMDS axes was defined as the model with the highest coefficient of determination (R-squared). Models were validated using two cross-validation approaches: A randomized 10-fold cross-validation and a spatial constrained 5-fold (Figure 1a). In each case, one fold was used as the validation set and the other folds as the calibration set. In the spatially constrained version, the inventory plots

were divided into five latitudinally non-overlapping folds in order to reduce the possible effects of spatial autocorrelation [66–70]. This can be expected to give more conservative estimates of model performance. Both the coefficient of determination ($R^2$) and the root-mean-square error (RMSE) are reported for model validation.

Many of the 27 predictor variables were strongly correlated. Therefore, we aimed to avoid overfitting by performing variable selection using a forward feature selection (ffs) method [66]. For the variable selection process, models with two predictors were first trained using all possible pairs of Landsat and environmental predictors. The pair of predictors that resulted in the best models were kept and additional predictors were iteratively added until they no longer significantly improved the model performance. The RF models using the selected Landsat and environmental variables were then applied over the whole Peruvian Amazonia in order to produce predictive maps of each of the main floristic gradients (NMDS axes 1–3). For the spatial predictions, the Landsat and environmental layers were rescaled to a spatial resolution of 450 m.

Finally, we classified the predictive maps into 10 classes using a k-means clustering method. The number of clusters in the k-means classification was based on an elbow method considering the ratio between-cluster and total variation. We selected the number of clusters in which the variation in the ratio between-cluster and total variation stabilized (Figure S1). To evaluate which genera were associated to each of the floristic classes, we performed an indicator analysis [71] based on indicator values at the genus level [72].

All analyses were carried out in R version 3.5.1 [73] using the packages "vegan" [74] for floristic ordinations, "indicspecies" [75] for indicator analysis, "raster" [76] for raster analysis and data extraction, "caret" [77] for machine learning model training and prediction, "CAST" [78] for cross-validation and forward feature selection, and "cluster" [79] for the k-means clustering.

## 3. Results

### 3.1. Floristic Patterns and Their Correlation with Landsat and Environmental Variables

We first visualized the spatial patterns of floristic relationships and floristic variation across gradients of the main response variable (floristic ordination axes NMDS 1, 2, and 3) at the genus level (Figure 2). Based on the floristic ordination (Figure 2a–c), overall, there was a floristic gradient over plots within hydromorphic forests, mountain forests, and lowland forests.

Floristic gradients were related both to the Landsat and to the environmental variables. The first axis of the floristic ordination (NMDS 1) was more strongly correlated with the Landsat variables (Table 1), particularly with the reflectance values of the near infrared spectrum (Landsat band 4, Figure 2d). The second and third axis (NMDS 2 and 3) were strongly correlated with the climatic variables (Table 1), particularly with the minimum temperature of the coldest month (bio6, Figure 2e) and precipitation seasonality (bio15, Figure 2f), respectively. The values of each of those variables correlated with NMDS 1, 2, and 3 varied spatially and according to each ecozone. For instance, the reflectance values of Landsat band 4 were higher in mountain forests (Figure 2g), bio6 values were higher in hydromorphic forests (Figure 2h), and bio15 values were smaller in hydromorphic forests (Figure 2i) compared to the other ecozones.

In the Mantel tests, dissimilarities in the tree genus composition were strongly correlated with differences in the values for all the Landsat variables, and, to a lesser degree, with differences in the environmental variables (Table 1). Floristic turnover was especially correlated with differences in the spectral values of the short-wave infrared bands (bands 5 and 7). In all cases, the strongest correlations with both the Landsat and environmental variables remained significant even when the effect of geographical distance was partialled out (Table 1). Floristic turnover was more strongly related with differences in the reflectance values of the Landsat bands than with differences in any of the environmental variables (bioclimatic layers, CEC, and elevation).

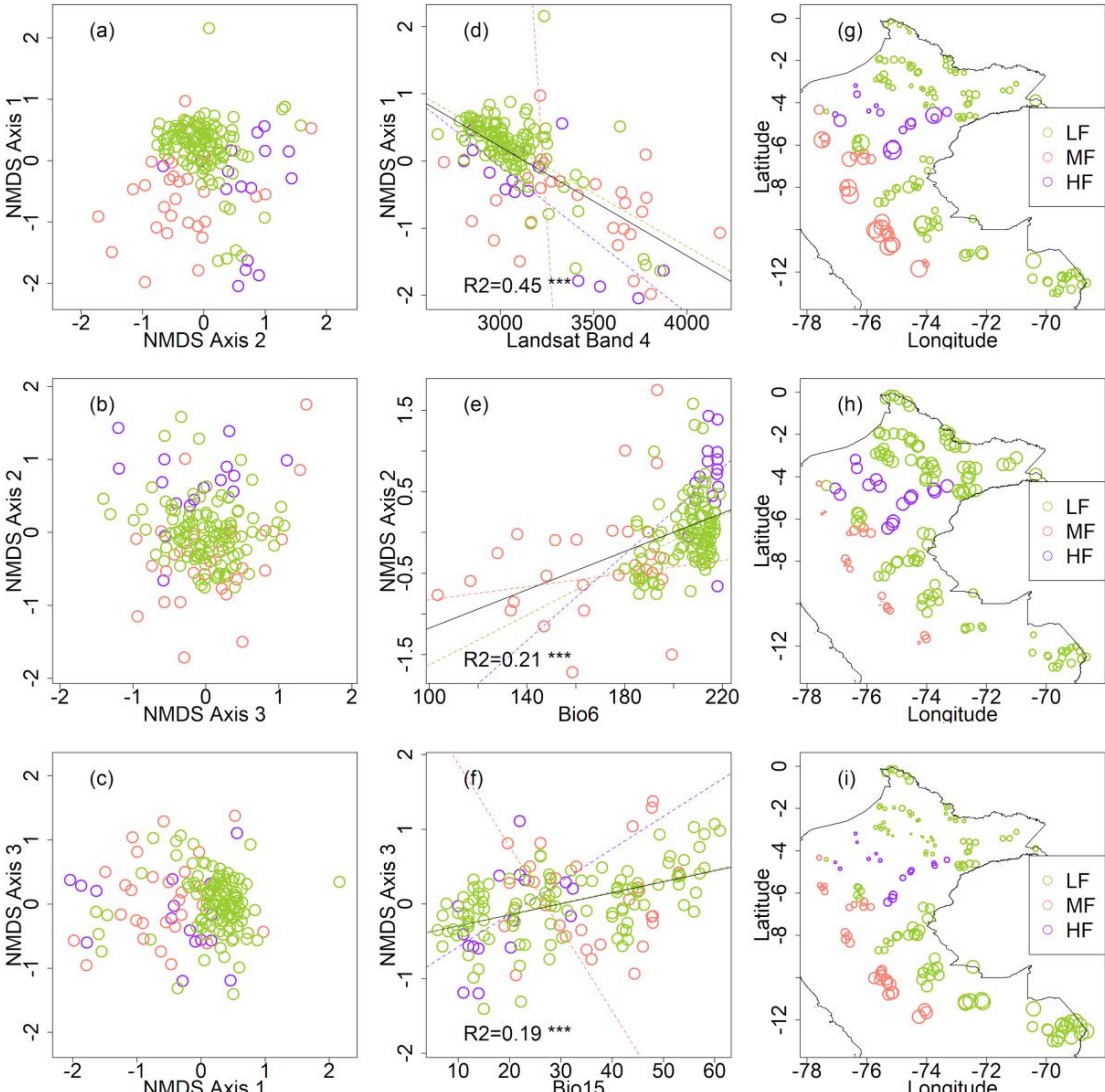

**Figure 2.** Floristic patterns across Peruvian Amazonia. (**a**–**c**) Non-metric multidimensional scaling (NMDS) ordination based on floristic dissimilarities between 157 plots (presence-absence data of trees, extended Sørensen dissimilarity). The three NMDS axes together capture 76% of the variation in the original compositional dissimilarity matrix with a stress of 0.19. (**d**–**f**). Relationship between each floristic NMDS axis and the variable with the strongest correlation with that axis (Bio6 = minimum temperature of the coldest month, Bio15 = precipitation seasonality). Regression lines were calculated for all data together (continuous line) or after subsetting the plots by ecozone (dashed lines). (**g**–**i**) map of the 157 tree inventory plots indicating the value of the corresponding best predictor (x axis from d, e, and f, respectively). In all panels, colors indicate the different ecozones: green for lowland forests (HF), orange for montane forests (MF), and purple for hydromorphic forests (HF). The full names and abbreviations of all variables are listed in Table S1.

**Table 1.** Relationships between the genus-level floristic composition of trees and environmental and Landsat variables. (1) Correlation values of partial Mantel tests between the distance matrices of the floristic composition and each of the Landsat and environmental variables separately, partialling out the correlation with the geographical distance. Statistical significance of each correlation coefficient was assessed with a Monte Carlo permutation test using 999 permutations. (2) Pearson correlation coefficients of simple linear correlation between each of the floristic ordination axes (NMDS 1–3) and each Landsat and environmental variable. *** $p < 0.001$, ** $P < 0.01$, * $p < 0.05$. Bands 1–7 refer to the Landsat bands, Bio1-19 refer to each of the bioclimatic variables (http://chelsa-climate.org/bioclim/), CEC refers to soil cation exchange capacity, and DEM refers to elevation. The full names and abbreviations of all variables are listed in Table S1.

| | Variables | Partial Mantel Test [1] | NMDS 1 [2] | NMDS 2 [2] | NMDS 3 [2] |
|---|---|---|---|---|---|
| Landsat | 1. Band 1 | 0.32 *** | 0.26 *** | 0 | 0.05 ** |
| | 2. Band 2 | 0.42 *** | 0.38 *** | 0.01 | 0.01 |
| | 3. Band 3 | 0.39 *** | 0.3 *** | 0.01 | 0.01 |
| | 4. Band 4 | 0.38 *** | 0.45 *** | 0 | 0.01 |
| | 5. Band 5 | 0.5 *** | 0.37 *** | 0 | 0.04 * |
| | 6. Band 7 | 0.5 *** | 0.38 *** | 0 | 0.05 ** |
| Environmental | 7. Bio1 | 0.26 *** | 0.11*** | 0.16 *** | 0 |
| | 8. Bio2 | 0.09 * | 0.15 *** | 0.11 *** | 0.07 *** |
| | 9. Bio3 | −0.12 | 0 | 0.04 * | 0.13 *** |
| | 10. Bio4 | 0 | 0.02 | 0.16 *** | 0.07 *** |
| | 11. Bio5 | 0.17 ** | 0.04 * | 0.07 ** | 0.03 * |
| | 12. Bio6 | 0.26 *** | 0.17 *** | 0.21 *** | 0.02 |
| | 13. Bio7 | 0.03 | 0.11 *** | 0.11 *** | 0.1 *** |
| | 14. Bio8 | 0.22 *** | 0.08 *** | 0.14 *** | 0 |
| | 15. Bio9 | 0.27 *** | 0.16 *** | 0.16 *** | 0.01 |
| | 16. Bio10 | 0.25 *** | 0.1 *** | 0.13 *** | 0 |
| | 17. Bio11 | 0.28 *** | 0.11 *** | 0.2 *** | 0 |
| | 18. Bio12 | 0.12 * | 0.18 *** | 0.01 | 0.09 *** |
| | 19. Bio13 | 0.04 | 0.13 *** | 0.05 * | 0.01 |
| | 20. Bio14 | −0.03 | 0.12 *** | 0 | 0.17 *** |
| | 21. Bio15 | −0.04 | 0.04 * | 0.02 | 0.19 *** |
| | 22. Bio16 | 0.04 | 0.12 *** | 0.06 * | 0.01 |
| | 23. Bio17 | −0.03 | 0.11 *** | 0 | 0.17 *** |
| | 24. Bio18 | 0.03 | 0.08 *** | 0.01 | 0.08 ** |
| | 25. Bio19 | −0.07 | 0.13 *** | 0 | 0.12 *** |
| | 26. CEC | 0.17 ** | 0.2 *** | 0.15 *** | 0.01 |
| | 27. DEM | 0.28 *** | 0.18 *** | 0.03 * | 0 |

## *3.2. Predicting Tree Community Composition at the Genus Level Throughout Peruvian Amazonia*

The performance of the random forest (RF) regression models predicting the NMDS ordination axes was consistently highest when the models included only variables selected in the feature forward selection process (Table 2). This was the case independently of which cross-validation method was used. The predictive power of the model based on Landsat variables was higher than that of the corresponding model based on environmental variables when modelling NMDS 1 but lower when modelling NMDS 2 or NMDS 3. Combining both Landsat and environmental variables in the RF models generally improved the model performance. In all models, the random cross-validation method derived higher $R^2$ and lower RMSE values than the spatial cross-validation (Table 2). The two cross-validation methods also led to different sets of predictor variables being selected for the final models.

**Table 2.** Statistical performance of the random forest models for predicting floristic ordination axes (NMDS 1–3 shown in Figure 2b,e,h). Models were compared by the type of cross-validation (CV) and by different initial sets of predictor variables. Final sets of predictor variables were selected based on the forward feature selection (ffs) method. Model performance was assessed through the root-mean-square-error (RMSE) and the coefficient of determination ($R^2$) averaged over all folds from the corresponding cross-validation. (*) The selected variables are listed using the same numbering as in Table 1.

| CV | Variable Combination | NMDS 1 | | NMDS 2 | | NMDS 3 | |
|---|---|---|---|---|---|---|---|
| | | ($R^2$) | RMSE | ($R^2$) | RMSE | ($R^2$) | RMSE |
| Random | Landsat | 0.531 | 0.455 | 0.219 | 0.467 | 0.199 | 0.467 |
| | Environmental (Env) | 0.372 | 0.537 | 0.449 | 0.414 | 0.321 | 0.414 |
| | Landsat + Env | 0.625 | 0.422 | 0.430 | 0.404 | 0.358 | 0.404 |
| | Selected by ffs | 0.626 | 0.411 | 0.605 | 0.395 | 0.396 | 0.395 |
| | Selected variables (*) | 5,8,13,16,24 | | 1,2,6,12,18,24,25,27 | | 1,12,24,27 | |
| Spatial | Landsat | 0.392 | 0.484 | 0.045 | 0.486 | 0.043 | 0.486 |
| | Environmental (Env) | 0.136 | 0.565 | 0.209 | 0.465 | 0.142 | 0.465 |
| | Landsat + Env | 0.483 | 0.437 | 0.201 | 0.431 | 0.166 | 0.431 |
| | Selected by ffs | 0.531 | 0.413 | 0.328 | 0.405 | 0.272 | 0.405 |
| | Selected variables (*) | 6,15,22,27 | | 2,12,18,22,27 | | 3,6,20,23 | |

Despite the differences in the final set of variables selected, the spatial predictions of the floristic ordination axes (NMDS 1–3) were similar between the RF analyses using the two different cross-validation methods (Figure 3). Their pixel-level correlations were 0.90 for NMDS 1 (Figure 3a,e), 0.95 for NMDS 2 (Figure 3b,f), and 0.90 for the NMDS axis 3 (Figure 3c,g). This indicates that the results are reasonably robust, and for further analyses, we simply selected the models with the highest coefficient of determination and lowest RMSE, i.e., those based on forward feature selection and random cross-validation. The maps obtained when combining the predictions for all three NMDS axes (Figure 3d,h) suggest clear differences in the predicted tree genus composition for different parts of Peruvian Amazonia.

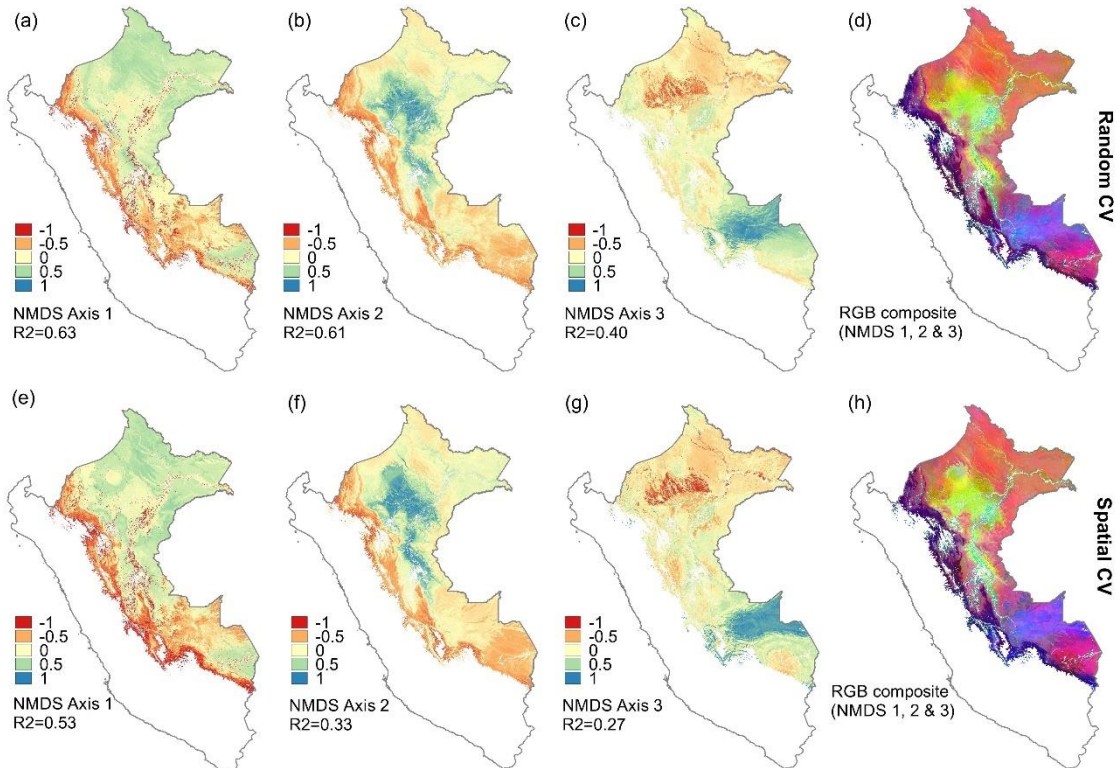

**Figure 3.** Spatial patterns in the community composition of tree genera across Peruvian Amazonia as represented by the predicted NMDS ordination scores. Predicted values were obtained with random forest regressions using a selection of Landsat and environmental variables (Table 1). The variable selection was based on forward feature selection and cross-validation was done using either 10 folds of randomly selected calibration data (**a–d**) or 5 spatially constrained folds (**e–h**). Panels a–c and e–g show predicted values for a single ban, and panels d and h the predicted values for all three NMDS axes assigned to red, green, and blue, respectively. Non-forest areas and areas above the 2000-m elevation were masked out and are shown in white.

On the basis of the predicted NMDS axes from the random cross-validation (Figure 3a–c), we classified Peruvian Amazonia into 10 classes using a k-means clustering method (Figure 4). On the basis of existing vegetation classifications [80–82], we could give coarse ecological and/or geographical interpretation to some of the classes (colors refer to those in Figure 4): Hydromorphic forests or "aguajales" in northern Peruvian Amazonia in the Pastaza fan (yellow), inundated forests (light green), mountain forests following the Andes (blue), lowland or "tierra firme" forest both in Northern Peru (red, gray) and Southern Peru (purple), and bamboo forests (orange).

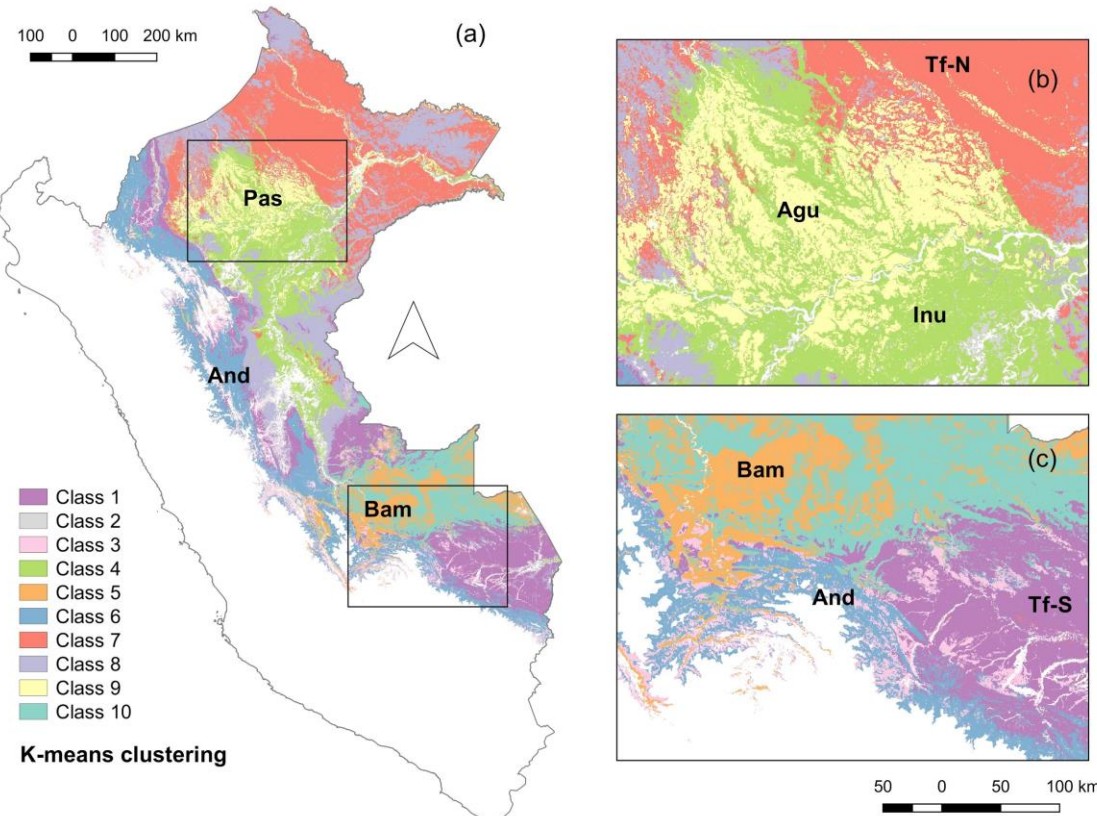

**Figure 4.** K-means clustering of Peruvian Amazonia into 10 classes based on the floristic ordination scores (NMDS axes 1-3) predicted using Landsat and environmental data with random forest regression. Non-forest areas and areas above the 2000-m elevation were masked out and shown in white (**a**). The letters on top of the maps refer to the Pastaza fan (Pas), mountain forests following the Andes (And), hydromorphic forests or "aguajales" (Agu), seasonally inundated forests (Inu), and tierra firme forests (Tf-N for Northern (**b**) and Tf-S for Southern (**c**)).

### 3.3. Indicator Analysis

On the basis of their coordinates, each of the 157 inventory plots was subsequently assigned to one of the 10 floristic classes shown in Figure 4. Indicator genus analysis was then carried out with the 463 genera observed in the plots. At least one indicator genus was found for most of the classes (Table 3). The exceptions were class 4 and 8 (shown in light green and light purple in Figure 4, respectively). Some classes had just a few indicator genera (class 2, 3, and 7) whereas others had more than 20 (class 10). Several known ecosystem classes were identified in our classification, such as hydromorphic forests (class 9), mountain forests (class 6), and bamboo-dominated forests (class 5). These were strongly associated with the genera *Mauritia*, *Cyathea*, and *Batocarpus*, respectively.

**Table 3.** Indicator analysis of 463 genera in relation to the classification of the floristic composition of trees in Peruvian Amazonia (see Figure 4).

| Class | Number of Inventory Plots | Number of Genera Associated | Genus | Indicator Value | *p*-Value |
|---|---|---|---|---|---|
| Class 1 | 22 | 5 | *Capirona* | 0.513 | 0.02 * |
| | | | *Quiina* | 0.503 | 0.017 * |
| | | | *Bertholletia* | 0.449 | 0.03 * |
| | | | *Pausandra* | 0.44 | 0.044 * |
| | | | *Aiouea* | 0.426 | 0.05 * |
| Class 2 | 7 | 1 | *Maclura* | 0.738 | 0.001 * |
| Class 3 | 11 | 2 | *Juglans* | 0.522 | 0.004 ** |
| | | | *Margaritaria* | 0.426 | 0.035 * |
| Class 4 | 13 | 0 | – | – | – |
| Class 5 | 8 | 10 | *Batocarpus* | 0.621 | 0.003 ** |
| | | | *Cavanillesia* | 0.604 | 0.002 ** |
| | | | *Copaifera* | 0.584 | 0.002 ** |
| | | | *Jacaratia* | 0.58 | 0.003 ** |
| | | | *Myriocarpa* | 0.561 | 0.005 ** |
| Class 6 | 7 | 6 | *Cyathea* | 0.703 | 0.001 * |
| | | | *Hieronyma* | 0.571 | 0.017 * |
| | | | *Palicourea* | 0.545 | 0.013 * |
| | | | *Chaunochiton* | 0.535 | 0.01 ** |
| | | | *Cinchona* | 0.526 | 0.008 ** |
| Class 7 | 43 | 2 | *Senefeldera* | 0.493 | 0.045 ** |
| | | | *Macoubea* | 0.463 | 0.041 * |
| Class 8 | 33 | 0 | – | – | – |
| Class 9 | 9 | 5 | *Mauritia* | 0.751 | 0.001 * |
| | | | *Pachira* | 0.711 | 0.024 * |
| | | | *Ferdinandusa* | 0.576 | 0.003 ** |
| | | | *Platycarpum* | 0.471 | 0.024 * |
| | | | *Diospyros* | 0.457 | 0.047 * |
| Class 10 | 4 | 22 | *Fusaea* | 0.7 | 0.001 * |
| | | | *Pseudobombax* | 0.678 | 0.003 ** |
| | | | *Quararibea* | 0.652 | 0.002 ** |
| | | | *Castilla* | 0.651 | 0.001 * |
| | | | *Ampelocera* | 0.633 | 0.004 ** |

## 4. Discussion

### 4.1. Correlates of Floristic Patterns

We found that the genus-level floristic patterns of trees across Peruvian Amazonia were strongly correlated with Landsat reflectance and, to a lesser extent, bioclimatic variables, elevation, and soil CEC as retrieved from the raster layers. A similar approach has been applied to map floristic patterns across all Amazonia but using fern and lycophyte species data to train the models [13]. Soil, climate, and Landsat layers had congruent roles in predicting the spatial patterns for both ferns at the species level and trees at the genus level, regardless of the two plant groups being phylogenetically remote, occupying different layers of the forest, and identified to different taxonomic resolutions.

Our findings are also in agreement with previous studies in which tree composition turnover and species distributions were found to be related with different environmental variables, such as soil, topography, and climate, in Amazonia and Central America [32–34,39,83,84]. We here expand previous findings by showing that genus-level floristic patterns of trees are also strongly related to reflectance Landsat values. This represents a major step towards floristic models since Landsat layers

have a broad spatial coverage and high resolution. Therefore, the same approach can also be applied for other plant groups and elsewhere than in Amazonia.

In earlier studies, the strongest floristic gradient has been related to soil properties in Amazonian forests, and canopy reflectance values have been found to predate there rather well [13,24,49]. On this basis, it seems likely that the first NMDS axis of our floristic ordination is related to soil characteristics. Even though soil digital maps have accuracy problems in poorly sampled areas in Amazonia [43], we found that changes in the soil cation exchange capacity (CEC) were correlated with the floristic turnover of tree genera but not as strongly as with changes in the canopy reflectance values derived from Landsat. Our results agree with previous findings of the soil characteristics being important factors explaining variation in the floristic composition of Amazonian forests. In agreement with an Amazon-wide study, we also observed that the second and third floristic ordination axes were more related to climate [13].

### 4.2. Predicting Floristic Composition of Trees Using Landsat Reflectance Values

The first floristic ordination axis (NMDS 1) was strongly correlated with the canopy reflectance values of all Landsat bands, and floristic turnover itself (extended Sørensen dissimilarity) with the corresponding differences in reflectance. This suggests that reflectance values can be used to spatially predict floristic patterns of trees over large areas in Amazonia, as previously suggested on the basis of observations concerning understory plants [13,24,25]. Our study with tree genera agrees with earlier studies on ferns and the shrub family Melastomataceae in that there is a tendency near infrared (NIR) and short-wave infrared (SWIR) Landsat bands (bands 4, 5, and 7) to be more strongly correlated with floristic variation than the visible bands (bands 1–3) [24–26,45,47]. As bands 4, 5, and 7 are assumed to reflect especially the biomass and moisture content of the vegetation, these features may be particularly strongly linked with the ecologically most relevant traits of the species in Amazonian forests.

Other remotely sensed data, such as airborne sensors, have been used to map the forest structure, carbon stocks, and functional and chemical traits in Amazonia [9–11,20,85,86], and they often have higher spatial and thematic resolution. Additionally, in other regions, higher spatial resolution remote sensing data has been used to map the floristic variation of tree species [87,88]. Nevertheless, Landsat-based data have a big advantage for many end users: They have global coverage and are freely available. In contrast, airborne sensor data are expensive, and since they are produced on demand, they only cover small areas and short time periods, which makes it difficult to replicate studies in other areas.

### 4.3. Interpretation of the Floristic Map and Indicator Genera

Our genus-level floristic map and the classification based on it (Figures 3d and 4a) show floristic patterns that can be interpreted on the basis of existing map products [80–82]. These include hydromorphic forests or "aguajales" in northern Peruvian Amazonia, mountain forests following the Andes, and bamboo forests in Southern Peruvian Amazonia. Two floristic classes appear within the Pastaza fan (Figure 4a), which we interpret as seasonally inundated forests (class 4) and hydromorphic forests (class 9). No genus was associated with class 4, but *Mauritia*, which is found in swamp areas in Amazonia [89–91], was strongly associated with floristic class 9. Another clear floristic pattern was found on the mountain forests near the Andes (class 6), which were strongly associated with the genus *Cyathea*, mainly distributed in the mountain regions of tropical America [92,93] and *Cinchona*, distributed in tropical Andean forests [94]. Class 5, which we related to bamboo forests, was strongly associated with the genus *Batocarpus*. This differs from earlier findings, which have related such forests to *Inga* and *Socratea* [95]. Tierra firme forests in Northern Peruvian Amazonia (class 7) were strongly associated with *Senefeldera*, and tierra firme forests in the South (class 1) were associated with *Bertholletia*, which within Peru is mainly distributed in Southern Amazonia [96,97]. Interestingly, many of the indicator genera strongly associated with our floristic classes contain hyperdominant tree species [98], such as *Bertholletia* in class 1, *Senefedera* in class 7, *Mauritia* and *Pachira* in class 9, and

*Pseudobombax* and *Quararibea* in class 10. Two of the indicator genera (*Bertholletia* and *Mauritia*) contain incipiently domesticated species but none have fully or semi-domesticated species [99].

Our classification agrees with previous ecosystem and vegetation classification maps in Peru [82,100], which also identify many of the ecosystems identified here, such as hydromorphic forests, lowland forests, bamboo-dominated forests, and tropical mountain forests. Our results also show some congruence with the forest trait diversity map of Peruvian Amazonia [11], where the forests were divided into six functional groups. Particularly, two functional groups geographically located in mountain and hydromorphic forests seem to agree with our respective floristic classes. The number of classes defined in our k-means clustering was based on the between-class and total variation ratio; nevertheless, different numbers of classes can also be analyzed for floristic congruency. Here, we found indicator genera associated with 8 out of the 10 floristic classes, which suggests strong floristic congruency of our proposed floristic division.

### 4.4. Practical Implications and Recommendations

Our floristic data were identified only to the genus level, so the results are not necessarily applicable to questions posed at the level of species. However, as earlier studies have shown that genus and species resolution data are highly correlated with each other [52,53], our results can be taken at least as the first approximation of patterns to be expected at the level of species. It is therefore reassuring, both for the Peruvian and for other Amazonian forest inventory efforts, that the inventory data can be used for meaningful floristic analyses even before the collections are fully identified.

Our results highlight the important contribution of combining field surveys and remote sensing towards improving the maps and classifications of floristic patterns in tropical forests. For the first time, the recent Peruvian national forest inventory data was used to predict broad-scale floristic patterns and derive a predicted community composition map for trees of Peruvian Amazonia. Using satellite imagery to expand insights for field inventories is particularly useful in data-poor areas but can be applied elsewhere as well. Identifying ecologically meaningful units of tropical areas is a crucial step towards better conservation strategies.

The environmental variables and Landsat data applied here are currently freely available, which enhances the possibilities of replicating the methodological framework for other biological groups and to other tropical regions. The forest inventory data used here consists of the first of a five-staged national forest inventory program. Therefore, more data will be available in the near future and our current floristic map can be further validated and improved. Floristic classes can also be used as a hypothesis of biogeographical units in Peruvian Amazonia and to assess the representativeness of conservation areas.

### 5. Conclusions

We found that the floristic patterns of trees at the genus level across Peruvian Amazonia are strongly correlated with reflectance values derived from Landsat imagery but also with other environmental variables, such as climate, soils, and elevation. In a floristic ordination, the first axis was strongly correlated with the spectral values of Landsat band whereas the secondary axes with climate data. By combining field data from the Peruvian national forest inventory, remotely sensed data, and machine learning, we derived a map that shows the predicted patterns in the community composition of trees at the genus level across Peruvian Amazonia. Comparison with earlier studies suggests the approach is robust and can be applied to other tropical regions, in order to reduce research gaps and to identify suitable areas for conservation.

**Supplementary Materials:** The following are available online at http://www.mdpi.com/2072-4292/12/10/1523/s1, Figure S1: Number of clusters for the k-means classification methods and their ratio between cluster and total variation (a) and the change between such ratio (b). Table S1: Variables used in the analyses.

**Author Contributions:** Conceptualization, P.P.C., G.Z., H.T. and K.R.; methodology, P.P.C.; formal analysis, P.P.C.; validation, P.P.C.; resources (Amazon Landsat mosaic), J.V.d.; resources (forest inventory data), E.G.R.; writing—original draft preparation, P.P.C.; writing—review and editing, P.P.C., G.Z., H.T., K.R., J.V.d., R.K. and E.G.R.; visualization, P.P.C. and G.Z. All authors have read and agreed to the published version of the manuscript.

**Funding:** This project was mainly funded by the University of Turku Graduate School (salary to P.P.C.) with contributions from the Academy of Finland (grant no. 273737 to H.T. and 296406 to R.K.) and the Independent Research Fund Denmark (salary of G.Z. from grant to Henrik Balslev).

**Acknowledgments:** We thank the Peruvian Forest and Wildlife Service (SERFOR) for kindly providing access to the forest inventory database and Mirkka M. Jones for inspiring discussions on the methodology.

**Conflicts of Interest:** The authors declare no conflict of interest. The funders had no role in the design of the study; in the collection, analyses, or interpretation of data; in the writing of the manuscript, or in the decision to publish the results.

**Data Availability:** The field data used in this study is property of the Peruvian Forest and Wildlife Service (SERFOR) and access was provided by the Peruvian government for this study only. Those interested in requesting access to the field data may contact SERFOR institution (see: www.serfor.gob.pe) at informes@serfor.gob.pe. The Landsat TM/ETM+ composite used in this research was deposited in IDA Research data storage service (www.fairdata.fi/en/ida/). Researchers interested in the Landsat-derived data should contact Hanna Tuomisto (hanna.tuomisto@utu.fi).

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
