# Peer review of "Mapping Floristic Patterns of Trees in Peruvian Amazonia Using Remote Sensing and Machine Learning"

_remotesensing, doi:10.3390/rs12091523_

Round 1

Reviewer 1 Report

Dear authors,

you did a great job and your manuscript is a high-quality manuscript. I recommend to publish the paper in Remote Sensing MDPI.

Best regards

Author Response

We appreciate the time and effort taken in reviewing our manuscript. We are glad that you are satisfied with our work.

Reviewer 2 Report

This study proposes a machine learning approach to map tree floristic patterns and distribution in Peruvian Amazonia using time series of Landsat data. The method includes supervised learning (random forest) and unsupervised learning (clustering). The authors did a fairly good job to explain the model design and the specification in detail. A couple of revisions are therefore recommended:

  1. The authors need to clarify the objectives of the paper. The goals are to map floristic variation using random forest or genu types using clustering, or both?
  2. In the classification practice, it is more common to use a machine-learning algorithm to directly predict genu types using satellite data. The authors need to elaborate more on why it is necessary to first predict NMDS first and then clustering the NMDS for final prediction.
  3. The NMDS is estimated by another model and served as key dependent variables, It would be necessary to have some description of the model. Does any of the biophysical variables are used when estimating of NMDS?
  4. The input data are all from different time periods. For example, the floristic data are from 2013-2018, satellite data are from 2000-2009, and BIOs are long term average from 1901 to 2016. The authors need to explain why the satellite data are from different time periods.
  5. For the feature selection part, it would be good to know why these bands and variables are more useful for the prediction compared to others. What is the theory behind?
  6. In the clustering part, how’s the cluster of 10 decided. Both elbow points and index-based approached are mentioned in the method section, but no evidence is shown in the results section. 
  7. The abstract needs to be re-written. It usually includes main findings and final results about the model such as accuracy assessment.

Author Response

  1. The authors need to clarify the objectives of the paper. The goals are to map floristic variation using random forest or genu types using clustering, or both?”

    Response: Both. We have rewritten both the Abstract and the Introduction and now explicitly state the aims in them.

  2. “In the classification practice, it is more common to use a machine-learning algorithm to directly predict genu types using satellite data. The authors need to elaborate more on why it is necessary to first predict NMDS first and then clustering the NMDS for final prediction”

    Response: Here we are interested in modelling floristic variation at the community level, not to build separate distribution models for the individual tree genera. In principle, community-level analyses could be achieved by stacking predictions made for the genera separately, but our data contains hundreds of different tree genera, many of which have low prevalence. In this situation, an ordination method can be used to summarize the multidimensional floristic variation into a small number of mutually orthogonal floristic axes. For this summarizing effort we use the NMDS technique. We have now added an explanation of this approach in the first paragraph under the subtitle “2.4. Data analyses”.

  3. “The NMDS is estimated by another model and served as key dependent variables, It would be necessary to have some description of the model. Does any of the biophysical variables are used when estimating of NMDS?”

    Response: NMDS is simply a technique to summarize and visualize the information from a floristic dissimilarity matrix, so none of the explanatory variables are used there. We now explain this more clearly in the first paragraph under the subtitle  “2.4. Data analyses”.

  4. "The input data are all from different time periods. For example, the floristic data are from 2013-2018, satellite data are from 2000-2009, and BIOs are long term average from 1901 to 2016. The authors need to explain why the satellite data are from different time periods”

    Response: This is the same response as the one given to the academic editor.

    In this article, we use data that are already available. The authors of the Landsat composite chose the 10-year period 2000-2009 because it maximized data availability: both Landsat 5 and Landsat 7 were operative in that period. It would be technically possible to produce a new Landsat composite for the field-data collection period (2013-2018), but the new composite would have a lower quality: Landsat 5 was no longer operative and Landsat 7 data suffered from scan line errors. For Landsat 8 data, the methods needed to produce this kind of radiometrically corrected data have not yet been developed. Given that the existing Landsat composite is based on representative reflectance values from a 10-year period, it can be assumed to represent relatively stable forest characteristics, just like the composition of canopy trees of >30 dbh does. Acknowledging the time gap, we nevertheless masked non-forest areas from the composite using an updated forest loss layer in the period 2001-2018. By doing so, we ensure that forested areas that were subject to major land use changes (e.g. converted to roads or pastures) after 2009 were eliminated from analysis. To clarify this logic, we have modified the first paragraph under the subtitle “2.3. Landsat and environmental data“.

  5. “For the feature selection part, it would be good to know why these bands and variables are more useful for the prediction compared to others. What is the theory behind?”

    Response: In the forward feature selection method (ffs), models with two predictors are first trained using all possible pairs of variables. The pair of predictors that derive the best models (higher coefficient of determination) are kept and additional predictors are iteratively added and kept if models are further improved. In this sense, the variables selected in this process are the ones that contribute and better explain the variation in the dependent variable (each of the floristic ordination axes NMDS 1, 2 and 3). We have added some explanation of what the most important Landsat bands reflect under the subtitle “4.2 Predicting floristic composition of trees using Landsat reflectance values”.

  6. “In the clustering part, how’s the cluster of 10 decided. Both elbow points and index-based approached are mentioned in the method section, but no evidence is shown in the results section”

    Response: The number of clusters in the k-means classification was based on an elbow method taking into account the ratio between-cluster and total variation. We selected the number of clusters in which the delta in the ratio between-cluster and total variation stabilized. We have briefly updated the description in the penultimate paragraph under the subtitle “2.4 Data analysis” and added a figure (Figure SM1) in the supplementary material section showing  two graphs g of the ratio between-clusters and total variation.

  7. “The abstract needs to be re-written. It usually includes main findings and final results about the model such as accuracy assessment”

    Response: Agreed, the abstract has been re-written.

Reviewer 3 Report

Title:

Monitoring herbaceous and shrub biomass in the Great Basin, USA using Landsat time series data

Summary:
The manuscript covers the timely and relevant topic of tropical forest biodiversity monitoring with a special focus on predicting tree genus diversity using Landsat satellite and environmental data. The authors integrate ground data from the National Forest Inventory (NFI) and extract the tree species variation at the genus level. The authors emphasize that they compiled the first such map for the Peruvian Amazon region. Even thought a cross-validation is used to validate the model, the quality of the map remains unclear as no map validation is done. The manuscript applies standard methods, is well-written and formatted and follows a classical structure of scientific manuscripts. Being a classical remote sensing application paper it fits well with the scopes of the journal „Remote Sensing“. The quality of the figures and tables is high.

Major comments:

1. The authors use a Landsat mosaic (obviously available from a former research project) using images from the years 2000-2009, whereas the NFI field data were collected between 2013-2018. Thus, in the worst case there is a time gap of 18 years between the image acquisition and the field data collection. As mentioned several times in the manuscript the advantage of using a freely available remote sensing data source is its availability. Thus, I cannot see why the authors accept to work with data having such a large time gap if images fitting the period of the NFI data collection are freely available. If there are no strong arguments against using fitting image dates I would recommend to do so.

2. The description of the NFI plot design is quite short and I would suggest to include a figure showing one plot, it's sub-plots and the buffer area of 450m used to extract the reflectance and environmental data. If i understood the text correctly, all subplots were treated as being independent. Thus, the buffers of the 7 subplots will largely overlap and then they cannot be treated independently. Also it would be relevant to provide some information on the accuracy of the GNSS positioning and the species / genus level identifications of trees. Often, the field teams have not the experiences in tropical species identifications.

3. This manuscript adds to the large amount of papers published in the last years that use a limited ground truthing dataset, apply a non-parametric model such as randomForest to a large set of remote sensing & GIS derived auxiliary variables to develop maps for very large areas. However, a validation of the maps based on independent ground data is often missing and thus the map quality (that is the accuracy at the pixel level predictions) is unknown but very likely lower than the presented model accuracies. This is particularly relevant for RandomForest models which tend to over fit. The presented cross-validation tests for for the accuracy and precision that the model reproduce the training data. Therefore, I strongly suggest to include a validation of the map to quantify the accuracy of the predictions. If no independent ground truth data of tree genus level for the time period is available a proper cross-validation of the map should be included. In the particular case of spatial predictions the cross-validation should consider the effect of spatial-autocorrelation- e.g. described in Meyer, H., Reudenbach, C., Wöllauer, S., Nauss, T., 2019. Importance of spatial predictor variable selection in machine learning applications – Moving from data reproduction to spatial prediction. Ecol. Modell. 411, 108815. doi:10.1016/j.ecolmodel.2019.108815

4. The abstract and some of the discussion and conclusion raise the expectation that a genus level tree species richness map is presented for the Peruvian Amazon. However, to my understanding a map with then classes which differ in their species composition is actually created. Using the indicator species approach some of the species that are likely associated with one of the classes is identified. However, this is not a genus level tree species maps as referred to in L25, L340-341,L389. I recommend to rephrase these paragraphs accordingly to highlight that NDMN groups are presented in the map.

Minor comments
Abstract:

L15 change "geographical" to "spatial"
L20 change "inventories" to "inventory"
L25 To my understanding this is not a genus level map (see major comment 4)
L29 As you use the NFI data and Landsat you have a lot of 'previous knowledge'

Introduction:

L40-L42 This is a strong assumption which needs some references as support
L67-L70 This is another strong assumption which needs some references as support
L84-L85 change to "..using freely available environmental layer, Landsat satellite imagery and NFI data..."

Materials and Methods:

L104 please provide a reference for MINAM (2016)
L117-L140 see major comment #2
L156 please include a table with the list of the 19 variables e.g. in the attachments
L185 The randomized 10-fold cross-validation does not consider the spatial autocorrelation in the data and this often overestimates the model accuracy. See major comment #3
L197-199 The elbow method is a quite subjective approach to determine the number of classes. If used a figure showing the 'elbow' should be included to support the decision for k=10

Results:

L213 remove "strong"

Discussion:

L340-L341 see major comment #4
L355-L358 I don't understand the intention of this statement. It is obvious that we expect differences in the species composition between different ecosystem types. Actually, this is one of the reasons why different types are designated.
L389 see major comment #4
L398 please explain how a map can mitigate research gaps
L399 please explain how the map can be used to identify suitable areas for management? What kind of management? Why should management follow broad categories of NDMS groups?

Figures and Tables:

I suggest to include a table with the 19 bio variables in the attachments

Author Response

  1. “The authors use a Landsat mosaic (obviously available from a former research project) using images from the years 2000-2009, whereas the NFI field data were collected between 2013-2018. Thus, in the worst case there is a time gap of 18 years between the image acquisition and the field data collection. As mentioned several times in the manuscript the advantage of using a freely available remote sensing data source is its availability. Thus, I cannot see why the authors accept to work with data having such a large time gap if images fitting the period of the NFI data collection are freely available. If there are no strong arguments against using fitting image dates I would recommend to do so”

    Response: This is the same response as the one given to the academic editor.

    In this article, we use data that are already available. The authors of the Landsat composite chose the 10-year period 2000-2009 because it maximized data availability: both Landsat 5 and Landsat 7 were operative in that period. It would be technically possible to produce a new Landsat composite for the field-data collection period (2013-2018), but the new composite would have a lower quality: Landsat 5 was no longer operative and Landsat 7 data suffered from scan line errors. For Landsat 8 data, the methods needed to produce this kind of radiometrically corrected data have not yet been developed. Given that the existing Landsat composite is based on representative reflectance values from a 10-year period, it can be assumed to represent relatively stable forest characteristics, just like the composition of canopy trees of >30 dbh does. Acknowledging the time gap, we nevertheless masked non-forest areas from the composite using an updated forest loss layer in the period 2001-2018. By doing so, we ensure that forested areas that were subject to major land use changes (e.g. converted to roads or pastures) after 2009 were eliminated from analysis. To clarify this logic, we have modified the first paragraph under the subtitle “2.3. Landsat and environmental data“.

  2. “The description of the NFI plot design is quite short and I would suggest to include a figure showing one plot, it's sub-plots and the buffer area of 450m used to extract the reflectance and environmental data. If i understood the text correctly, all subplots were treated as being independent. Thus, the buffers of the 7 subplots will largely overlap and then they cannot be treated independently. Also it would be relevant to provide some information on the accuracy of the GNSS positioning and the species / genus level identifications of trees. Often, the field teams have not the experiences in tropical species identifications”

    Response: We have updated Figure 1 in the manuscript so that it includes the two different schemes of the inventory plots. Our analyses were done at the resolution of plots, not subplots. We have clarified the sampling methods under the subtitle “2.2 Floristic data”.

  3. “This manuscript adds to the large amount of papers published in the last years that use a limited ground truthing dataset, apply a non-parametric model such as randomForest to a large set of remote sensing & GIS derived auxiliary variables to develop maps for very large areas. However, a validation of the maps based on independent ground data is often missing and thus the map quality (that is the accuracy at the pixel level predictions) is unknown but very likely lower than the presented model accuracies. This is particularly relevant for RandomForest models which tend to over fit. The presented cross-validation tests for for the accuracy and precision that the model reproduce the training data. Therefore, I strongly suggest to include a validation of the map to quantify the accuracy of the predictions. If no independent ground truth data of tree genus level for the time period is available a proper cross-validation of the map should be included. In the particular case of spatial predictions the cross-validation should consider the effect of spatial-autocorrelation- e.g. described in Meyer, H., Reudenbach, C., Wöllauer, S., Nauss, T., 2019. Importance of spatial predictor variable selection in machine learning applications – Moving from data reproduction to spatial prediction. Ecol. Modell. 411, 108815. doi:10.1016/j.ecolmodel.2019.108815”

    Response: We have now added a spatial cross-validation for model validation in order to avoid possible effects of spatial-autocorrelation. We have included additional references for such purpose (Robert et al. 2017, Meyer et al. 2019, Schratz et al. 2019, Karasiak et al., 2019). This is explained in the third and fourth paragraph under the subtitle “2.4 Data analyses” in the methods section. Additionally, we updated Figure 1 to show the spatial arrangement of the spatially constrained cross-validation. The new validation results (coefficient of determination and root-mean-square error) are now added in Table 2, so that model performances are easy to see. Finally, we visually compared both sets of predictions and also ran a Pearson correlation between the predicted floristic ordination axes of both models. These additions made the results section somewhat longer, as seen in the first two paragraphs under the subtitle “3.2 Predicting tree community composition at genus-level throughout Peruvian Amazonia”, but also gave us increased confidence on the robustness of the results.

  4. “The abstract and some of the discussion and conclusion raise the expectation that a genus level tree species richness map is presented for the Peruvian Amazon. However, to my understanding a map with then classes which differ in their species composition is actually created. Using the indicator species approach some of the species that are likely associated with one of the classes is identified. However, this is not a genus level tree species maps as referred to in L25, L340-341,L389. I recommend to rephrase these paragraphs accordingly to highlight that NDMN groups are presented in the map”

    Response: All the analyses are done at the taxonomic resolution of genus, and we have edited the text to clarify this. We have also checked that the text no longer discusses species or genus richness, as the analyses are indeed focused on turnover rather than richness.  

Minor comments

  • L15 change "geographical" to "spatial"

Response: The Abstract was rewritten and the change was made accordingly.

  • L20 change "inventories" to "inventory"

Response: The Abstract was rewritten and the change was made accordingly.

  • L25 To my understanding this is not a genus level map (see major comment 4)

Response: We have reworded several sentences in the updated version of the manuscript, including the rewritten Abstract, to clarify that the genus-level tree community composition map is based on the predicted floristic ordination axes.

  • L29 As you use the NFI data and Landsat you have a lot of 'previous knowledge'

Response: The Abstract was rewritten.

  • L40-L42 This is a strong assumption which needs some references as support

Response: We have added several references that support the general idea that the combination of field-data and remote sensing data can be used to predict biological patterns over large areas.

  • L67-L70 This is another strong assumption which needs some references as support

Response: The supporting references were actually listed in the first sentence of the paragraph. We have now rewritten the paragraph so as to better convey our logic. We also added one more reference to support our claim. 

  • L84-L85 change to "..using freely available environmental layer, Landsat satellite imagery and NFI data..."

Response: The Introduction was restructured accordingly.

  • L104 please provide a reference for MINAM (2016)

Response: We have added the reference for MINAM (2016)

  • L117-L140 see major comment #2

Response: We have added a schematic representation of the spatial structure of each inventory plot on Figure 1.

  • L156 please include a table with the list of the 19 variables e.g. in the attachments

Response: We have added a table (Table SM1) under the supporting material section listing all the variables, their names and abbreviations.

  • L185 The randomized 10-fold cross-validation does not consider the spatial autocorrelation in the data and this often overestimates the model accuracy. See major comment #3

Response: As explained in the response of major comment 3, we have included a spatially constrained cross-validation which takes into consideration the effect of autocorrelation.

  • L197-199 The elbow method is a quite subjective approach to determine the number of classes. If used a figure showing the 'elbow' should be included to support the decision for k=10

Response: We have updated the description under the subtitle “2.5 Data analyses” and added a figure (Figure SM1) in the supplementary materials section showing two graphs of the ratio between-clusters and total variation.

  • L213 remove "strong"

Response:  Changed accordingly.

  • L340-L341 see major comment #4

Response: In the updated manuscript we are referring to community composition of trees at genus-level.

  • L355-L358 I don't understand the intention of this statement. It is obvious that we expect differences in the species composition between different ecosystem types. Actually, this is one of the reasons why different types are designated.

Response: Our point here is that we can, on the basis of pre-existing knowledge, interpret some of the identified floristic classes. We have now reformulated the sentence and added the relevant references under the second paragraph of the subtitle “4.3. Interpretation of the floristic map and indicator genera”. 

  • L389 see major comment #4

Response: In the updated manuscript we are referring to community composition of trees at genus-level highlighting that the map is based on the predicted floristic ordination axes.

  • L398 please explain how a map can mitigate research gaps

Response: We have changed “mitigate” and “reduce”, and by such, we mean that we are able to build predictions of floristic patterns in areas where field data is missing.

  • L399 please explain how the map can be used to identify suitable areas for management? What kind of management? Why should management follow broad categories of NDMS groups?

Response: We have removed the management part in the phrase and kept only the conservation part.

  • I suggest to include a table with the 19 bio variables in the attachments

Response: We have added a table under the supporting material’ section (Table SM1) listing all the variables, their codes and names.

Round 2

Reviewer 3 Report

With the revised version of the manuscript the authors present a considerably improved version. They were able to address all my major concerns and provided logical and sound justifications. The integration of the additional spatially constraint cross-validation significantly adds to the contribution that this manuscript can make. I would suggest to use only the results from the spatially constrained cross-validation to reduce the complexity of the analysis. However, since the authors show that the derived maps are quite similar I would leave the decision with the authors.